# The Mitochondrial Ca^2+^ Overload via Voltage-Gated Ca^2+^ Entry Contributes to an Anti-Melanoma Effect of Diallyl Trisulfide

**DOI:** 10.3390/ijms21020491

**Published:** 2020-01-13

**Authors:** Chinatsu Nakagawa, Manami Suzuki-Karasaki, Miki Suzuki-Karasaki, Toyoko Ochiai, Yoshihiro Suzuki-Karasaki

**Affiliations:** 1Department of Dermatology, Nihon University Hospital, Tokyo 101-830, Japan; chinatsu.y0503@gmail.com (C.N.); toochiai@j.com.home.net.jp (T.O.); 2Plasma ChemiBio Laboratory, Nasushiobara, Tochigi 329-2813, Japan; ksmanami181@gmail.com (M.S.-K.); ksmiki181@gmail.com (M.S.-K.)

**Keywords:** diallyl trisulfide, melittin, melanoma, apoptosis, mitochondrial Ca^2+^, store-operated Ca^2+^ channel, acidification, Ca^2+^ channel blocker, voltage-gated Ca^2+^ channel

## Abstract

*Allium* vegetables such as garlic (*Allium sativum* L.) are rich in organosulfur compounds that prevent human chronic diseases, including cancer. Of these, diallyl trisulfide (DATS) exhibits anticancer effects against a variety of tumors, including malignant melanoma. Although previous studies have shown that DATS increases intracellular calcium (Ca^2+^) in different cancer cell types, the role of Ca^2+^ in the anticancer effect is obscure. In the present study, we investigated the Ca^2+^ pathways involved in the anti-melanoma effect. We used melittin, the bee venom that can activate a store-operated Ca^2+^ entry (SOCE) and apoptosis, as a reference. DATS increased apoptosis in human melanoma cell lines in a Ca^2+^-dependent manner. It also induced mitochondrial Ca^2+^ (Ca^2+^_mit_) overload through intracellular and extracellular Ca^2+^ fluxes independently of SOCE. Strikingly, acidification augmented Ca^2+^_mit_ overload, and Ca^2+^ channel blockers reduced the effect more significantly under acidic pH conditions. On the contrary, acidification mitigated SOCE and Ca^2+^_mit_ overload caused by melittin. Finally, Ca^2+^ channel blockers entirely inhibited the anti-melanoma effect of DATS. Our findings suggest that DATS explicitly evokes Ca^2+^_mit_ overload via a non-SOCE, thereby displaying the anti-melanoma effect.

## 1. Introduction

*Allium* vegetables such as garlic (*Allium sativum* L.) are rich in allyl sulfides that have been shown to prevent human chronic diseases, including cancer [1]. L-alliin (S-allyl-l-cysteine sulfoxide) is the major allyl sulfide component in garlic, which is converted into 2-propensulfenic acid by the endogenous enzyme alliinase, thereby producing the unstable thiosulfinate compound allicin (*S*-allyl-2-prop-2-ene-thiosulfinate). Allicin readily breaks down to form organosulfur compounds (OSCs), including allyl sulfides such as diallyl trisulfide (DATS), diallyl disulfide (DADS), and diallyl sulfide (DAS), as well as ajoene and vinyl dithiins [2]. OSCs can modulate the immune system and attenuate inflammation, and these effects contribute to cancer prevention [3]. DATS, DADS, and DAS are known to prevent the well-characterized chemical-induced skin carcinogenesis [4,5,6]. Moreover, the OSCs have anticancer activity in a variety of cancer cell types, including skin cancer such as melanoma and basal cell carcinoma [7]. Of these, DATS is the most widely studied. This compound induces apoptosis in a range of cancer cells, including malignant melanoma, osteosarcoma, and leukemia [8,9,10,11]. DATS also potentiates apoptosis induced by tumor necrosis factor-related apoptosis-inducing ligand (TRAIL) in human prostate cancer and melanoma cells [12,13,14]. The induction and amplification of apoptosis are associated with activation of caspases, upregulation of multiple pro-apoptotic molecules, including death receptor (DR)4/DR5, Bax, and Bak as well as downregulation of anti-apoptotic molecules such as Bcl-2 and Bcl-xL [10]. Fully activation of TRAIL-induced apoptosis in melanoma cells requires not only the intrinsic mitochondrial pathway but also the endoplasmic reticulum (ER) stress pathway [14]. The observations are consistent with the clinical findings that TRAIL can activate robust intrinsic death signaling but are ineffective in drug-resistant cancers, including malignant melanoma [15,16]. Strikingly, DADS and DATS exhibit the anticancer and TRAIL-sensitizing effects in a tumor-selective manner [8,14], and they commonly induce the intracellular generation of reactive oxygen species (ROS) [8,9]. These observations support the view that oxidative stress plays a vital role in the cancer cell-selective killing and TRAIL sensitization [17,18,19].

Ca^2+^ is a highly versatile intracellular second messenger that regulates numerous complicated cellular processes, including cell activation, proliferation, and death. The increase in the cytosolic Ca^2+^ concentration ([Ca^2+^]_cyt_) results in the activation of Ca^2+^/calpain, an intracellular Ca^2+^-dependent cysteine protease, thereby resulting in the processing of the mitochondrial localized pro-apoptotic molecule, apoptosis-inducing factor, and caspase-independent apoptosis. The activation of Ca^2+^/calpain occurs during the apoptosis of glioblastoma and neuroblastoma induced by DAS, DADS, and DATS [8,9]. Meanwhile, an excess rise in the mitochondrial Ca^2+^ concentration ([Ca^2+^]_mit_) leads to increased permeability of the mitochondrial membrane, mitochondrial dysfunction, and release of pro-apoptotic molecules such as cytochrome c and Apaf-1, resulting in caspase-dependent apoptosis. Store-operated Ca^2+^ entry (SOCE) is the principal route by which external Ca^2+^ enters in non-excitable cells. Ca^2+^ depletion in the endoplasmic reticulum (ER) triggers SOCE. The depletion induces the translocation of stromal interaction molecules 1 (STIM1) to ER/plasma membrane (PM) junctional regions, where the molecule activates the PM channel ORAI1, leading to Ca^2+^ entry from the extracellular milieu. Recently, Gualdani and colleagues reported that SOCE mediates cisplatin-induced Ca^2+^_mit_ overload, ROS production, and apoptosis [20].

Melittin, a 26-amino acid amphipathic peptide from bee venom, exhibits the anticancer effect against a variety of tumor cells, including melanoma, leukemia, osteosarcoma, lung, and bladder cancer cells. Melittin induces apoptosis in these tumor cells in a caspase-dependent manner [21,22,23]. Moreover, earlier studies have suggested that melittin induces apoptosis through Ca^2+^ entry [23,24]. Notably, melittin is also known to activate SOCE via the activation of Ca^2+^-independent phospholipase A_2_ (iPLA_2_) [25,26]. Collectively, by analogy with cisplatin, it is plausible to speculate that melittin exhibits the anticancer effect through Ca^2+^ overload via SOCE. While DATS preferentially acts on tumor cells [8,14], melittin acts non-specifically, as cisplatin does. Therefore, the identification of the Ca^2+^ transport pathways preferentially activated by DATS may help to define the Ca^2+^ pathways targeted in future tumor-selective therapy.

In the present study, we explored the ability of DATS to cause Ca^2+^ dysregulation and its possible role in the anti-melanoma effect using melittin as a reference. We found that DATS explicitly induces massive Ca^2+^ dysregulation and apoptosis through a voltage-gated Ca^2+^ entry (VGCE) pathway that is distinct from SOCE.

## 2. Results

### 2.1. DATS Reduces Cell Viability and Increases Apoptosis in a Ca^2+^- and Caspase-Dependent Manner

We treated human malignant melanoma cell lines A375 and A2058 with varying concentrations of DATS for 72 h and measured cell viability by using the WST-8 assay. Based on our previous study, we determined the range of the concentrations examined [14]. DATS (≥100 μM) decreased the viability of A375 cells in a dose-dependent manner (Figure 1A). DATS also reduced the viability of A2058 in a dose-dependent manner (Figure 1B). However, cellular sensitivity to DATS varied considerably in different experiments. As a result, the IC_50_ values for A375 and A2058 cells were 159 ± 32 and 228 ± 60 μM, respectively (the mean ± SD, *n* = 3–6). Flow cytometric analyses using annexin V and PI staining revealed that 72-h-treatment with DATS (100 μM) alone resulted in a massive increase in apoptotic (annexin V-positive) cells. TRAIL markedly augmented the effect while the pan-caspase inhibitor Z-VAD-FMK entirely blocked it (Figure 1C,D). We found that Ca^2+^ was a critical regulator of drug sensitivity. Treatment with the extracellular Ca^2+^ chelator EGTA (≤0.5 mM) or the intracellular Ca^2+^ chelator BAPTA had minimal effect on cell viability. However, these chelators significantly reduced the anticancer effect of DATS in A375 and A2058 cells (Figure 1E,F).

### 2.2. Melittin Exhibits Anti-Melanoma Effect in a Ca^2+^-Dependent Manner

Melittin is known to be a potent inducer of apoptosis in melanoma cells. Consistent with this view, treatment with the compound (≥2.5 μg/mL) for 72 h resulted in a robust increase in apoptotic (annexin V-positive) cells in A375 cells (Figure 2A). Meanwhile, the treatment minimally increased necrotic (annexin V-negative) cells. The extracellular Ca^2+^ removal by EGTA (0.5 mM) augmented the effect of the subtoxic dose (1 μg/mL) of melittin. On the other hand, it mitigated the increase in apoptosis while enhancing the increase in necrosis caused by the toxic concentration (5 μg/mL) of melittin (Figure 2B).

### 2.3. DATS Increases [Ca^2+^]_mit_ without Increasing [Ca^2+^]_cyt_

Next, we determined whether DATS affected the intracellular Ca^2+^ level. First, we tested the effect on [Ca^2+^]_cyt_. We used the Ca^2+^-ATPase inhibitor, thapsigargin (Tg), as a positive control, because it depletes the ER Ca^2+^ stores, thereby stimulating SOCE. Tg substantially increased [Ca^2+^]_cyt_, while DATS at the concentration of up to 200 μM minimally increased it (Figure 3A,B). On the other hand, DATS increased [Ca^2+^]_mit_ in a dose-dependent manner (Figure 3C,D). This increase occurred immediately, developed over time, and the effect of DATS (200 μM) was even higher than that induced by the positive control A23187 (5 μM).

### 2.4. Melittin Increases Both [Ca^2+^]_cyt_ and [Ca^2+^]_mit_ in an Extracellular Ca^2+^-Dependent Manner

The site-specific Ca^2+^ measurements showed that melittin (1.3–5 μg/mL) increased both [Ca^2+^]_cyt_ and [Ca^2+^]_mit_ in a dose-dependent manner (Figure 4A,C). Removal of the extracellular Ca^2+^ entirely abolished the increase in [Ca^2+^]_cyt_ but partially reduced the increase in [Ca^2+^]_mit_, indicating that these effects depend on the extracellular Ca^2+^ influx to different degrees (Figure 4B,D).

### 2.5. Acidification Has a Reciprocal Effect on Ca^2+^_mit_ Overload Caused by DATS and Melittin

The microenvironments, including the extracellular pH conditions, contribute to the various aspects of malignant phenotypes of cancer cells [27,28]. Therefore, we examined the effect of altered extracellular pH on Ca^2+^_mit_ overload. Cells were suspended in HBSS with different pH 6.8 and 7.4, treated with DATS, and measured for the intracellular Ca^2+^ levels. Notably, after DATS treatment, a small but significant increase in [Ca^2+^]_cyt_ was seen at pH 6.8, but not pH 7.4 (Figure 5A,B). Similarly, a greater extent of Ca^2+^_mit_ overload occurred at pH 6.8 compared to pH 7.4, and this impact was more pronounced for the higher concentration of the compound (Figure 5C,D). The effect was apparent for higher concentrations of DATS. Next, we analyzed the effect of the extracellular Ca^2+^ removal on Ca^2+^_mit_ overload seen under different pH. Cells were suspended in HBSS (pH 7.4 or 6.8) containing CaCl_2_ or EGTA and treated with DATS. At pH 7.4, the Ca^2+^ removal almost entirely abolished the increase in [Ca^2+^]_mit_, while at pH 6.8, it reduced the increase only a modestly (Figure 5E,F). In contrast to DATS, acidification significantly mitigated Ca^2+^_mit_ overload caused by melittin. A much smaller increase in [Ca^2+^]_mit_ occurred at pH 6.8 compared to at pH 7.4 in response to a range of concentrations of melittin (1.3–5 μg/mL). Figure 5G shows the data for melittin (2.5 μg/mL). Notably, we observed a comparable level of [Ca^2+^]_mit_ increase in the cells placed in HBSS (pH 7.4) and Ca^2+^-containing buffer. The [Ca^2+^]_mit_ rise seen in HBSS (pH 6.8) and Ca^2+^-free buffer (pH 7.4) after melittin treatment were also comparable (Figure 5G,H).

### 2.6. Acidification Mitigates SOCE

The data presented so far indicated that acidification reduced Ca^2+^_mitt_ overload caused by melittin. Since melittin is known as a potent activator of SOCE, we determined the effect of acidification on SOCE. We activated SOCE by the Ca^2+^ depletion-Ca^2+^ re-addition technique and examined the impact of acidification on the Ca^2+^ entry. At neutral extracellular pH, the addition of Tg to cells in a Ca^2+^-free medium resulted in a small and transient increase in [Ca^2+^]_cyt_, indicating the Ca^2+^ release from the ER. The re-addition of Ca^2+^ to Tg-treated cells led to a higher and persistent increase in [Ca^2+^]_cyt_, and the inositol-1,4,5-triphosphate receptor (IP_3_R)/SOCE antagonists 2-aminoethoxydiphenyl borate (2-APB) entirely blocked this Ca^2+^ response, verifying the onset of SOCE (Figure 6A). Even without any Ca^2+^ depletion in the ER (i.e., in the absence of Tg), a smaller but significant [Ca^2+^]_cyt_ rise was seen following Ca^2+^ re-addition and was minimally affected by 2-APB (Figure 6A, Right), indicating the onset of a store-independent Ca^2+^ influx (non-SOCE). Meanwhile, at pH 6.8, no substantial SOCE was observed, while the non-SOCE was increased by the acidification (Figure 6B).

### 2.7. Ca^2+^ Channel Blockers Inhibit Ca^2+^_mit_ Overload and Cell Death Caused by DATS

To gain insight into the transport pathway(s) involved Ca^2+^_mitt_ overload caused by DATS, we look for Ca^2+^ channel antagonists affecting the [Ca^2+^]_mit_ rise. Eventually, we found that the Ca^2+^ channel blockers nifedipine, verapamil, and diltiazem significantly inhibited the effect of DATS. The inhibitory effect was seen more pronouncedly at pH 6.8 than at pH 7.4 (Figure 7A). Under the acidic conditions, nifedipine exhibited the most potent effect (70% reduction), while the other compounds inhibited it around 50% (Figure 7A). 2-APB had an inhibitory effect comparable to verapamil and diltiazem. Meanwhile, at the neutral pH conditions, these inhibitors mitigated the [Ca^2+^]_mit_ rise only modestly (20%; Figure 7A). Next, we examined the effect of Ca^2+^ blockers on the anti-melanoma effect in highly growing cells. Nifedipine, verapamil, and diltiazem significantly inhibited the anti-melanoma effect of DATS (200 μM; Figure 7B).

## 3. Discussion

In this study, we elucidated the possible role of Ca^2+^ dysregulation in the anti-melanoma effect of DATS with a particular interest in the Ca^2+^ pathways involved. We found that DATS induced caspase-dependent apoptosis in melanoma cells, although relatively high concentrations of DATS are required (Figure 1). These results are consistent with previous studies, including our own, which demonstrate that DATS induces apoptosis in a variety of cancer cell types [8,9,10,11,14]. We noticed that cell confluency considerably affected the sensitivity to the compound. Cells growing near confluent tended to be more susceptible than those growing at low confluency. In agreement with the previous reports by others [23,24], melittin also primarily induced apoptosis in the cells (Figure 2A). Moreover, we found that extracellular Ca^2+^ entry was necessary for apoptosis caused by DATS and melittin (Figure 1E,F and Figure 2B). The Ca^2+^_mit_ uptake is generally due to so-called reservoir function of the organelles: the mitochondria take up Ca^2+^ in response to increased [Ca^2+^]_cyt_ thereby preventing excessive [Ca^2+^]_cyt_ rise while they release Ca^2+^ in response to decreased [Ca^2+^]_cyt_ thereby counteracting excessive [Ca^2+^]_cyt_ depletion. Notably, we found that DATS and melittin had different impacts on [Ca^2+^]_cyt_ and [Ca^2+^]_mit_. DATS substantially increased [Ca^2+^]_mit_, but not [Ca^2+^]_cyt_ (Figure 3), suggesting that the Ca^2+^_mit_ overload is a result of the activation of a specific Ca^2+^ transport pathway rather than a simple Ca^2+^_mit_ uptake. Meanwhile, melittin markedly increased [Ca^2+^]_cyt_ and [Ca^2+^]_mit_ (Figure 4). However, extracellular Ca^2+^ removal entirely abolished the former, while attenuated the latter partially (Figure 4B,D). These results suggest that the [Ca^2+^]_cyt_ rise primarily results from extracellular Ca^2+^ entry while the [Ca^2+^]_mit_ increase is due to Ca^2+^ transport of both extracellular and intracellular Ca^2+^. Dysregulation of [Ca^2+^]_mit_ is a master cause of cell death. A fine-tuned increase in [Ca^2+^]_mit_ supports energy metabolism, cell activation, and cell survival, while Ca^2+^_mitt_ overload can damage mitochondrial integrity, thereby inducing mitochondrial permeability transition pore (MPTP) opening and the resulting release of apoptogenic proteins [28,29,30]. Collectively, we conclude that different Ca^2+^ transport pathways participate in the Ca^2+^_mitt_ overload and apoptosis caused by DATS and melittin (Figure 8).

Strikingly, DATS seems to evoke Ca^2+^_mit_ overload through different Ca^2+^ pathways, depending on the extracellular pH. At pH 7.4, DATS exhibits the effect mainly via the extracellular Ca^2+^ entry (Figure 5E), while at pH 6.8, DATS seemed to use both extracellular and intracellular Ca^2^ entry pathways (Figure 5F). Ca^2+^ released from the IP_3_R can easily reach the mitochondrial matrix via the IP_3_R–voltage-dependent anion channel (VDAC1)–mitochondrial Ca^2+^ uniporter (MCU) pathway [31,32]. The ER is the primary mechanism for regulating [Ca^2+^]_mit_ and tethered to mitochondria via a mitochondria-associated membrane; the physical association allows rapid Ca^2+^ transport through specified microdomains [33,34]. The IP_3_R physically links to VDAC1 on the outer mitochondrial membrane and transports Ca^2+^ into the mitochondrial matrix [35,36]. We previously showed that Ca^2+^_mit_ remodeling in osteosarcoma cells was under the control of Ca^2+^ uptake through the MCU [37]. [Ca^2+^]_mit_ is also regulated by Ca^2+^_mit_ efflux through several different pathways, including the mitochondrial Na^+^/Ca^2+^ exchanger (NCLX), Ca^2+^/H^+^ antiporter, and MPTP [38,39,40,41,42]. Notably, our previous study showed that the NCLX inhibitor CGP-37157 causes Ca^2+^_mit_ overload and apoptosis in osteosarcoma cells [43]. Therefore, increased Ca^2+^ transport via the IP_3_R–VDAC1–MCU pathway and decreased Ca^2+^_mitt_ extrusion could contribute to the Ca^2+^_mitt_ overload caused by DATS.

SOCE is the most ubiquitous pathway for Ca^2+^ transport from the extracellular space and activated following the depletion of Ca^2+^ stores from the ER. Recently, SOCE has emerged as critical machinery for Ca^2+^ influx, contributing to various malignant phenotypes in cancer cells [44,45,46]. SOCE is also shown to be responsible for cisplatin-induced apoptosis [20]. It is noteworthy that melittin can activate iPLA_2_ to produce arachidonic acid and lysophospholipids and activates SOCE in vascular smooth muscle cells [25,26]. Given that melittin activates SOCE in melanoma cells, too, this action can expect to cause massive Ca^2+^ dysregulation, as cisplatin does. A notable property of SOCE is its sensitivity to acidification; SOCE is suppressed by decreasing extracellular pH in several cell types [44,45]. In agreement with these reports, acidification entirely abolished SOCE in melanoma cells (Figure 6). Notably, the Ca^2+^_mit_ overload induced by melittin was mainly dependent upon extracellular Ca^2+^ entry and was suppressed by acidification (Figure 5). Extracellular Ca^2+^ removal and acidification caused a similar degree of reduction, suggesting that extracellular Ca^2+^ entry is specifically sensitive to acidification, as observed with SOCE. Together, our findings strongly suggest that melittin evokes Ca^2+^_mit_ overload via SOCE activation. This view is consistent with other reports demonstrating the requirement of SOCE for active Ca^2+^_mit_ uptake [47,48]. However, further studies, including the examination of the involvement of SOCE components such as STIM1 and ORAI1, are necessary for validation.

Another important finding in the present study is that Ca^2+^ channel blockers significantly mitigate Ca^2+^_mit_ overload and the anti-melanoma effect of DATS at acidic pH (Figure 7). These findings suggest a role of VGCE in Ca^2+^_mit_ overload and apoptosis. Notably, acidification increased Ca^2+^_mit_ overload caused by DATS (Figure 5), and sensitivity to Ca^2+^ channel blockers was higher at acidic pH than at neutral pH (Figure 7A). The latter finding seems to provide another evidence for the role of VGCE because Kato and colleagues have shown that an acidic pH (5.4–6.5) increases Ca^2+^ influx in mouse B16 melanoma cells through VGCE [49]. The activation of VGCE requires plasma membrane depolarization, which leads to the loss of the negative membrane potential, the driving force transporting extracellular Ca^2+^ into inner cells. Accordingly, SOCE could be compromised under these conditions. Thus, acidification activates VGCE while mitigates SOCE, thereby possibly playing a vital role in switching the Ca^2+^ entry pathway from SOCE to VGCE. The microenvironments of tumor cells are acidic due to an H^+^ efflux caused by increased vacuolar ATPase activity, and this acidification causes multiple malignant phenotypes, including increased proliferation, drug resistance, and metastasis [27,28]. Therefore, our observations that high confluent cells are relatively high responders of DATS could be explained by the predominant role of VGCE under acidic extracellular microenvironments, because the extracellular pH drops in cells growing at high density.

There remain issues to be resolved in future studies. In particular, the identification of voltage-gated Ca^2+^ channels (VGCCs) involved in Ca^2+^_mit_ overload caused by DATS is challenging. Our preliminary experiments revealed that melanoma cells expressed Ca_v_1.2 and Ca_v_1.3 isoform of l-type VGCCs. Notably, the previous studies demonstrate the occurrence of the coordinate control of STIM1-ORAI1 and Ca_v_1.2 [50,51]. Therefore, there is an intriguing possibility that Ca_v_1.2 plays a role in Ca^2+^_mit_ overload via a specific interaction with SOCE. Further studies are necessary to clarify the molecular entity of VGCCs involved and to test this hypothesis.

In conclusion, the present study demonstrated that DATS and melittin exhibit the anti-melanoma effects by evoking pro-apoptotic Ca^2+^_mit_ overload through the different pathways, possibly VGCE and SOCE. Our findings suggest the involvement of multiple Ca^2+^ entry pathways in Ca^2+^_mit_ remodeling in melanoma cells and switching between them, depending on apoptotic stimuli and pH microenvironments.

## 4. Materials and Methods

### 4.1. Materials

All chemicals were purchased from Sigma Aldrich (St. Louis, MO, USA) unless otherwise specified. Alliin and allicin were obtained from Cayman Chemical Co. (Ann Arbor, MI, USA). Soluble recombinant human TRAIL was obtained from Enzo Life Sciences (San Diego, CA, USA). The pan-caspase inhibitor Z-VAD-FMK was purchased from Merck Millipore (Darmstadt, Germany). All insoluble reagents were dissolved in dimethylsulfoxide (DMSO) and diluted with high glucose-containing Dulbecco’s modified Eagle’s medium (DMEM) supplemented with 10% fetal bovine serum (FBS) or Hank’s balanced salt solution (HBSS; pH 7.4, Nissui Pharmaceutical Co., Ltd., Tokyo, Japan; final DMSO concentration, <0.1%) prior to use.

### 4.2. Cell Culture

Cells were cultured in 10% FBS/DMEM supplemented with 100 U/mL penicillin and 100 μg streptomycin (Pen-Strep, Thermo Fisher Scientific Japan, Tokyo, Japan) in a 95% air/5% CO_2_ humidified atmosphere unless otherwise specified. The human melanoma cell lines A375 (cell number CRL-1619) and A2058 (cell number IFO 50276) were obtained from the American Type Culture Collection (ATCC, Manassas, VA, USA) and the Japanese Collection of Research Bioresources (JCRB) Cell Bank of National Institutes of Biomedical Innovation, Health, and Nutrition (Osaka, Japan), respectively. Cells were harvested by incubation with 0.25% trypsin-ethylenediaminetetraacetic acid (EGTA; Thermo Fisher Scientific, Waltham, MA, USA) for 5 min at 37 °C.

### 4.3. Cell Viability Assay

Cell viability was measured by WST-8 assay using Cell Counting Reagent SF (Nacalai Tesque, Inc., Kyoto, Japan) as previously described [43] with modifications. This method is a colorimetric assay based on the formation of a water-soluble formazan product. Briefly, cells were seeded at a density of 8 × 10^3^ cells/well in 96-well plates (Corning Incorporated, Corning, NY, USA) and cultured with agents to be tested for 72 h at 37 °C before the addition of 10 μL of cell counting reagent SF and further incubation for 2 h. The absorbances were measured at 450 nm using an ARVO MX microplate reader (PerkinElmer Japan Co., Ltd., Yokohama, Japan).

### 4.4. Cell Death Assay

Cell death was quantitatively assessed by double-staining with fluorescein isothiocyanate (FITC)-conjugated annexin V and propidium iodide (PI) as previously described [37]. Briefly, cells plated in 24-well plates (2 × 10^5^ cells/well) were treated with the agents to be tested for 20 h and stained with FITC-conjugated annexin V and PI using a commercially available kit (Annexin V FITC Apoptosis Detection Kit I: BD Biosciences, San Jose, CA, USA). The stained cells (10,000 cells) were analyzed in FL-1 and FL-2 channel of a FACSCalibur flow cytometer (BD Biosciences) using the CellQuest software (BD Biosciences). Four cellular subpopulations were evaluated: viable cells (annexin V-negative and PI-negative); early apoptotic cells (annexin V-positive and PI-negative); late apoptotic cells (annexin V-positive and PI-positive); and necrotic/membrane-damaged cells (annexin V-negative and PI-positive).

### 4.5. Intracellular Ca^2+^ Measurements

Changes in Ca^2+^_cyt_ and Ca^2+^_mit_ levels were measured using Fluo4-AM and rhod2-AM (Dojindo Kumamoto, Japan), respectively, as previously described [37]. For improvement of mitochondrial localization of rhod 2-AM [52], it was reduced to the colorless, nonfluorescent dihydrorhod 2-AM by sodium borohydride, according to the manufacturer’s protocol. Cells were loaded with 4 μM each of Fluo 4-AM or dihydrorhod 2-AM for 40 min at 37 °C, washed with HBSS. Then, the cells (1 × 10^6^/mL) were resuspended in HBSS in 96-well plates. The cells were manually added with the agents to be tested. Then, the cells were measured for fluorescence in a microplate reader (Fluoroskan Ascent, ThermoFisher Scientific) with excitation and emission at 485 and 538 nm (for Fluo 4-AM), respectively, and 542 and 592 nm (for rhod 2-AM), respectively. For the analysis of Ca^2+^ release and SOCE, Fluo4-AM-loaded cells were suspended in a Ca^2+^-free medium (HBSS supplemented with 1 mM EGTA) and treated with 2 μM Tg for 10 min and then added to 2 mM CaCl_2_. For analysis of non-SOCE, cells in the Ca^2+^-free medium were treated with the medium for 10 min and then added to 2 mM CaCl_2_.

### 4.6. Statistical Analysis

Data are presented as mean ± standard deviation (SD) or standard error (SE) and were analyzed by one-way analysis of variance followed by Tukey’s post hoc test using add-in software for Excel 2016 for Windows (SSRI, Tokyo, Japan)*. p* < 0.05 was considered statistically significant.

## Figures and Tables

**Figure 1 ijms-21-00491-f001:**
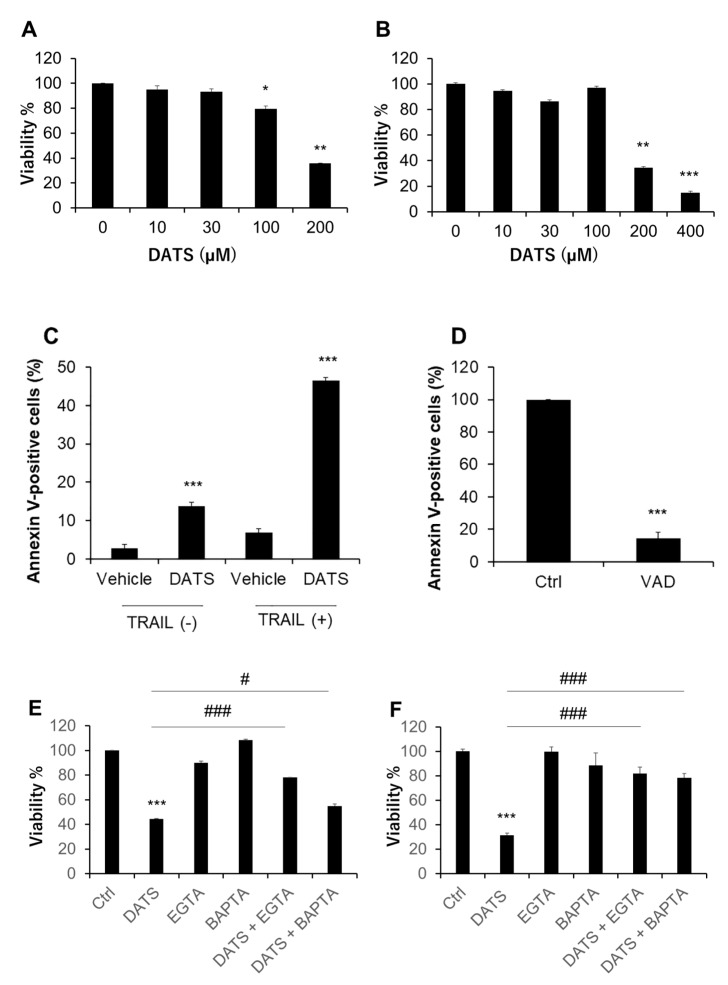
DATS exhibits the anti-melanoma effect in a caspase- and Ca^2+^-dependent manner. (**A**) A375 and (**B**) A2058 cells in Dulbecco’s modified Eagle’s medium supplemented with 10% fetal calf serum (FCS/DMEM) were treated with the indicated concentrations of DATS for 72 h and analyzed for viability using the WST-8 assay. * *p* < 0.05; ** *p* < 0.01 vs. untreated control. (**C**) A375 cells were treated with DATS (100 μM) in the absence or presence of TRAIL (100 ng/mL) for 72 h, stained with FITC-conjugated annexin V and propidium iodide (PI), and analyzed in a flow cytometer. *** *p* < 0.001 vs. untreated control. (**D**) A375 cells were treated with DATS (200 μM) in the absence (Ctrl) or presence of Z-VAD-FMK (10 μM; VAD) for 72 h and processed as described above. *** *p* < 0.001 vs. DATS alone. (**E**,**F**) Effect of Ca^2+^ removal on the anti-melanoma effect. (**E**) A375 and (**F**) A2058 cells were treated with DATS (200 μM) in the absence or presence of EGTA (0.2 mM) or BAPTA (30 μM) for 72 h and analyzed for viability using the WST-8 assay. **** p* < 0.01 vs. untreated control. # *p* < 0.05; ### *p* < 0.001 vs. DATS alone. Data represent the mean ± SD (*n* = 3–6).

**Figure 2 ijms-21-00491-f002:**
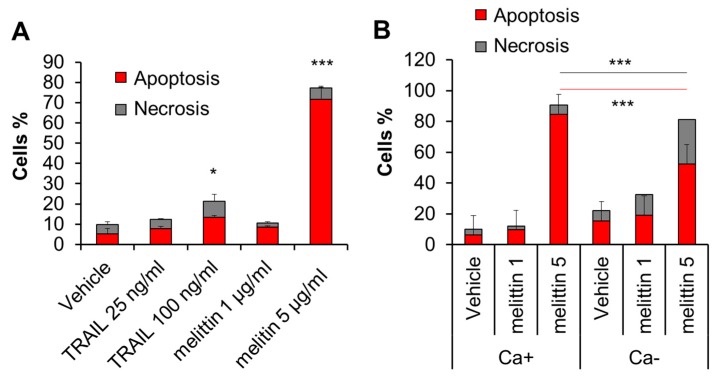
Melittin exhibits anti-melanoma effect in a Ca^2+^-dependent manner. (**A**) A375 cells in FCS/DMEM were treated with the recombinant human TRAIL (25, 100 ng/mL) or melittin (1 or 5 μg/mL) alone for 72 h. (**B**) The cells were treated with melittin (1 or 5 μg/mL) in the absence (Ca+) or presence of EGTA (0.5 mM) (Ca−) for 72 h. The cells were evaluated for cell death modality as described in the legend of Figure 1. Data represent the mean ± SD (*n* = 3). * *p* < 0.05; *** *p* < 0.001.

**Figure 3 ijms-21-00491-f003:**
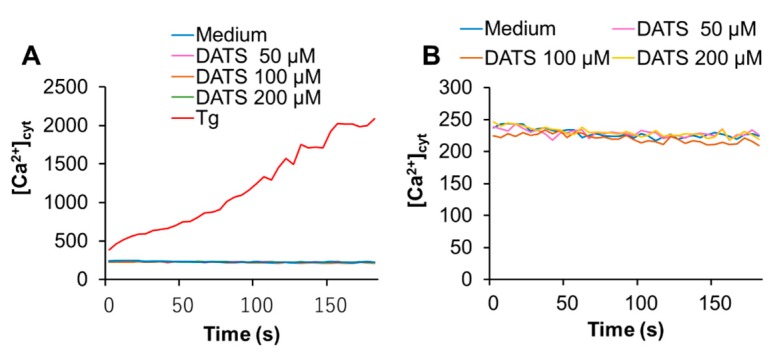
DATS induces Ca^2+^_mit_ overload with minimal Ca^2+^_cyt_ increase in melanoma cells. (**A**,**B**) DATS has minimal effect on [Ca^2+^]_cyt_. A375 cells were loaded with Fluo4-AM, added with DATS (50, 100, and 200 μM) or Tg (1 μM), and immediately measured for fluorescence for 3 min with excitation and emission at 485 and 538 nm, respectively, in a microplate fluorescence reader. [Ca^2+^]_cyt_ was evaluated as described in Materials and Methods. (**C**,**D**) DATS induces Ca^2+^_mit_ overload in a dose-dependent manner. The cells were loaded with dihydrorhod 2-AM, treated with DATS as described above**,** and measured for fluorescence with excitation and emission at 542 and 592 nm, respectively. Data are shown as *F/F_0_*_,_ where *F* and *F_0_* represent the fluorescence of the sample and control (medium-treated cells), respectively. Data shown in (**D**) represent the mean ± SE (*n* = 3). **** p* < 0.001 vs. untreated control.

**Figure 4 ijms-21-00491-f004:**
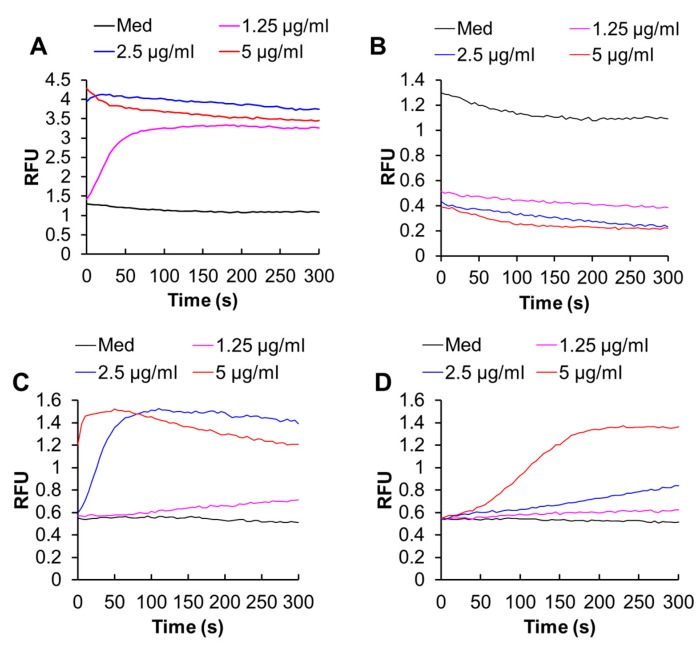
Melittin increases the intracellular Ca^2+^ levels in melanoma cells. (**A**,**B**) A375 cells were loaded with Fluo4-AM**,** suspended in HBSS containing 1 mM CaCl_2_ (**A**) or 1 mM EGTA (**B**), and added with the indicated concentrations of melittin, and measured for fluorescence with excitation and emission at 485 and 538 nm, respectively in a microplate fluorescence reader. The vertical line shows relative fluorescence units (RFU). (**C**,**D**) The cells were loaded with dihydrorhod-2-AM**,** suspended in HBSS containing 1 mM CaCl_2_ (**C**) or 1 mM EGTA (**D**), and added with the indicated concentrations of melittin, and measured for fluorescence with excitation and emission at 542 and 592 nm, respectively in a microplate fluorescence reader.

**Figure 5 ijms-21-00491-f005:**
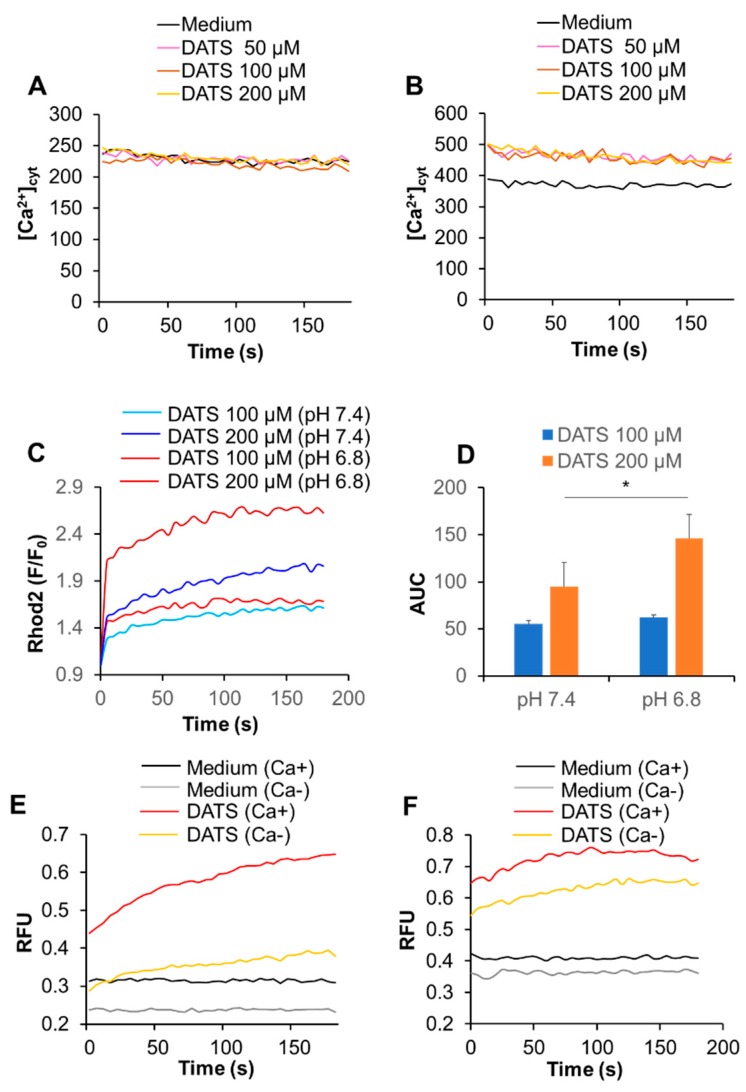
Acidification has a reciprocal effect on Ca^2+^_mit_ overload caused by DATS and melittin. (**A**,**B**) A375 cells were suspended in HBSS (pH 7.4) (**A**) and 6.8 (**B**), treated with the indicated concentrations of DATS, and measured for fluorescence with excitation and emission at 485 and 538 nm, respectively in a microplate fluorescence reader. (**C**,**D**) Dihydrorhod2-AM-loaded A375 cells were suspended in HBSS (pH 7.4 or 6.8), treated with DATS (100 and 200 μM) and measured for fluorescence with excitation and emission at 542 and 592 nm, respectively, in a microplate fluorescence reader. Data are shown as F/F_0_, where F and F_0_ represent the fluorescence of the sample and control, respectively (*n* = 3–5). The data shown in (**D**) represent mean ± SD of the area under the curve (AUC) for 3 min (*n* = 3). (**E**,**F**) The cells were suspended in HBSS, pH 7.4 (**E**) or 6.8 (**F**) containing 1 mM CaCl_2_ (Ca+) or 1 mM EGTA (Ca−), treated with DATS (200 μM), and measured for the fluorescence as described above. Data are shown as relative fluorescence unit (RFU). (**G**,**H**) The cells were suspended in HBSS (pH 7.4 or 6.8) (**G**) or HBSS (pH 7.4) containing 1 mM CaCl_2_ (Ca+) or 1 mM EGTA (Ca−) (**H**), treated with melittin (2.5 μg/mL) and measured for the fluorescence. * *p* < 0.05.

**Figure 6 ijms-21-00491-f006:**
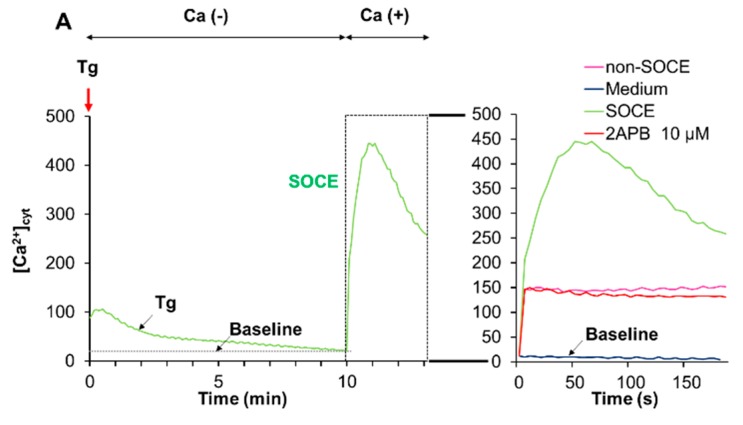
Acidification mitigates store-operated Ca^2+^ entry (SOCE). Fluo4-AM-loaded A375 cells were suspended in a Ca^2+^-free HBSS supplemented with 1 mM EGTA at pH 7.4 (**A**) or pH 6.8 (**B**), added with thapsigargin (Tg, 2 μM, red arrow) and incubated for 10 min to deplete intracellular Ca^2+^ stores. Then, two mM Ca^2+^ was added to the cells (SOCE). After the addition of Tg, the fluorescence was monitored with excitation and emission at 485 and 538 nm, respectively. For validating SOCE, the cells in the Ca^2+^-free buffer were treated as described above in the presence of 2-APB (10 μM). For non-SOCE measurement, the cells in the Ca^2+^-free buffer were treated with the medium and then added with 2 mM Ca^2+^ (non-SOCE). Note that at pH 7.4, a higher increase in [Ca^2+^]_cyt_ was seen in the SOCE trace compared to the non-SOCE trace, and this effect was entirely abolished by 2-APB. Meanwhile, no such effect was seen at pH 6.8.

**Figure 7 ijms-21-00491-f007:**
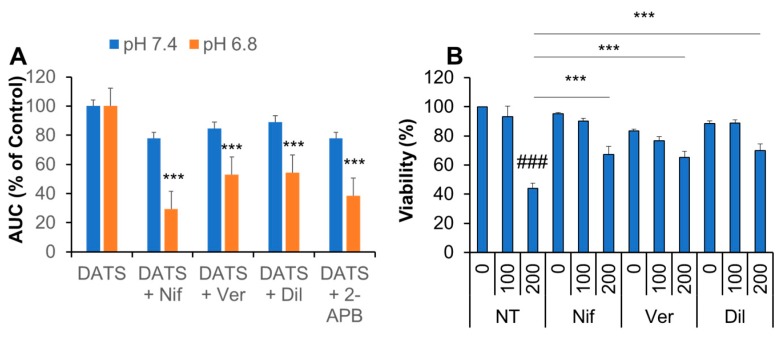
Ca^2+^ blockers inhibit Ca^2+^_mit_ overload and the anti-melanoma effect by DATS. (**A**) Dihydrorhod2-AM-loaded A375 cells suspended in HBSS, pH 6.8 were added with DATS (200 μM) in the absence or presence of 1 μM each of nifedipine (Nif), verapamil (Ver), diltiazem (Dil), or 2-APB (10 μM) and measured for fluorescence with excitation and emission at 542 and 592 nm, respectively. The data represent mean ± SD of the area under the curve (AUC) for 3 min. The values of DATS alone are set as 100%. **** p* < 0.001 vs. DATS alone (*n* = 3) (**B**) The cells were treated with DATS (100 and 200 μM) in the absence or presence of each inhibitor as described above for 72 h and measured for cell viability by the WST-8 assay. Data represent the mean ± SD (*n* = 3). *### p* < 0.001 vs. untreated control set as 100%. **** p* < 0.001 vs. DATS alone.

**Figure 8 ijms-21-00491-f008:**
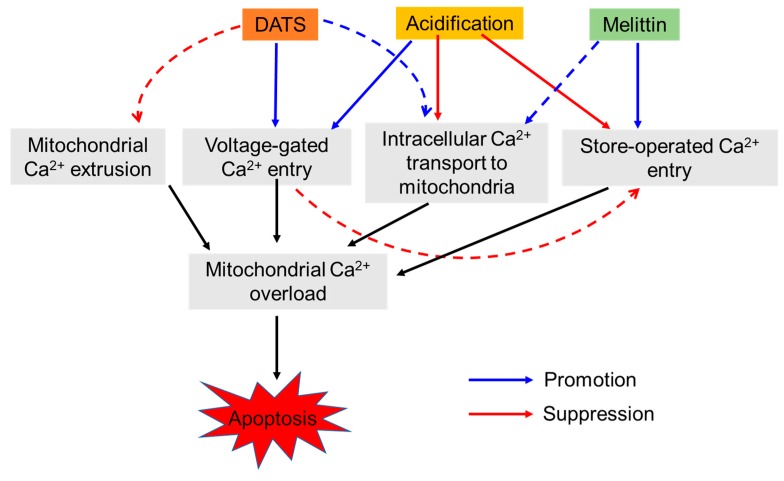
A schematic summary of the present study. Both DATS and melittin induce mitochondrial Ca^2+^ overload and apoptosis in melanoma cells. However, different pathways seem to contribute to the effects, depending on the apoptotic stimuli and the extracellular pH. Specifically, at neutral pH, the extracellular Ca^2+^ entry is predominant. Melittin primarily uses store-operated entry (SOCE), while DATS mainly utilizes yet unknown extracellular Ca^2+^ entry pathway possibly including voltage-gated Ca^2+^ entry (VGCE). Under acidic pH, SOCE is mitigated. In turn, intracellular Ca^2+^ entry plays a substantial role in the effects of melittin and DATS. Increased Ca^2+^ transport from the intracellular stores (e.g., the endoplasmic reticulum) to the mitochondria and decreased mitochondrial Ca^2+^ extrusion could contribute to the event. Under the acidic conditions, DATS, but not melittin, seems to utilize VGCE as the primary route for Ca^2+^ entry, leading to mitochondrial Ca^2+^ overload and apoptosis. The VGCE could exacerbate SOCE inactivation, and Ca^2+^ dysregulation due to the loss of the negative membrane potential, the driving force transporting extracellular Ca^2+^ into inner cells through SOCE.

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
