# Peer review of "The Mitochondrial Ca^2+^ Overload via Voltage-Gated Ca^2+^ Entry Contributes to an Anti-Melanoma Effect of Diallyl Trisulfide"

_ijms, 2020, doi:10.3390/ijms21020491_

Round 1
Reviewer 1 Report
In this study, Nakagawa et al., demonstrated that Ca2+ dysregulation via a Ca2+ channel blocker-sensitive channel playing a critical role in the anti-melanoma effect of DATS.
The topic of the research is evaluated and described in a clarify manner and according to me, the manuscript can be considered acceptable for the publications.
Author Response
We want to thank the Reviewers for thoughtfully and critically reviewing our manuscript. All recommendations and suggestions by you are acknowledged and appreciated.
We revised the title and rearranged the text, including Introduction, Results, and Discussion, to make the background, the aim of this study, and the conclusion drawn more clearly.
For authors who are not experts in this field, we added the chemical background of garlic-derived organosulfur compounds and their cancer-preventive and anticancer effects in the Introduction. We revised the reference according to the rearrangement.
Also, throughout the manuscript, we carefully rechecked the English style and revised sentences more concisely and clearly.
All revision is indicated by underline
Also, we attached Figures at a resolution of 300 dpi.
Reviewer 2 Report
Previous major concerns have been addressed.
Author Response
We want to thank the Reviewers for thoughtfully and critically reviewing our manuscript. All recommendations and suggestions by you are acknowledged and appreciated.
We revised the title and rearranged the text, including Introduction, Results, and Discussion, to make the background, the aim of this study, and the conclusion drawn more clearly.
For authors who are not experts in this field, we added the chemical background of garlic-derived organosulfur compounds and their cancer-preventive and anticancer effects in the Introduction. We revised the reference according to the rearrangement.
Also, throughout the manuscript, we carefully rechecked the English style and revised sentences more concisely and clearly.
All revision is indicated by underline
Also, we attached Figures at a resolution of 300 dpi.
This manuscript is a resubmission of an earlier submission. The following is a list of the peer review reports and author responses from that submission.
Round 1
Reviewer 1 Report
The paper by Nakagawa and coworkers focused on the effects of DATS and allicin in melanoma cells in order to clarify the mechanisms of cell death. The ability of DATS to induce apoptotic cell death in melanoma cells is already published; however the manuscript introduces new data especially about the involvement of calcium channels.
Some points should be clarified and revised before publication:
1) To better compare the cytotoxic effects of DATS, Allicin and alliin, authors should performed viability test using more drug concentration in order to evaluate for each compound IC50.
Moreover it is not clear in figure 2 where alliin was used. Authors should clarify.
2) in Annexin/PI analysis, used in Figure 3, dot plot shoul be shown in order to appreciate the percentage of cells that are Ann-PI-, Ann+PI+, Ann+PI- and Ann-PI+. Authors declare that PI positive cells are minimal buti t is not evident in Figure 2.
3) Authors declare to compare the effects of the three OSCs in non transformed cells. DATS and allicin are shown in the figure 4 but alliin is not present. Authors should revise.
4) The effects induced by DATS impair calcium flux and it has been demonstrated that Diallyl sulfides activate cation channel belonging to the TRP family as TRPA1 and TRPV1. Moreover 2-APB is able to block TRP channel. Authors should investigate the role or TRP channels in DATS induced apoptotic cell death using potent and specific antagonists.
Reviewer 2 Report
Nakagawa et al described in a clarify manner the evidences about the role of mitochondrial Ca2+ deregulation and OSCs treatments into melanoma cell lines.
Despite,the anti-melanoma effects of OSCs in different tumor cell types are knows, in this study the authors investigated in preliminary manner the involvement of DATS and mellitin in apoptosis pathway regulation by different modulation of Ca2+ and its trafficking.
Apotosis modulation was investigated by different approaches. The introduction and discussion sections are write in a clear manner. In my opinion , the manuscript require to check the English language and style and to check figure and figure legends.
Reviewer 3 Report
This manuscript studied the effects of three forms of OSCs in melanoma cells. Major concerns include:
In figure 2, is the Calcium effect also observed in A2058 cells? Why did EGTA treatment promote cell death at 100 µM, but inhibit cell death at 200 µM in figure 2E? The rationale for using a combination of TRAIL and OSC used in figure 3 is not clear. It is not clear how authors present results in figure 3. Figure 3B indicates that 72 hours of DATS treatment induced 20-50% cell death, while 3C and 3D showed that DATS treatment for 72 hours induced 100% cell death. How experiments in figure 4 were performed? Was calcium included or not? Higher doses of DATS is required as figure 2C also indicated that 100 µM did not have an effect on A2058 cells either. It will be useful to compared conditions with or without calcium. Figure 4C presented negative results. Moreover, the rationale for using a combination for TRAIL and OSC is not clear. The mechanisms underlying different effects of DATS and Allicin observed in figure 5 are not discussed? What is the physiological relevance of using different pH conditions in figure 6? Proper controls to demonstrate the activation and inactivation of SOCE pathway need to be included in experiments presented in figure 7.